# A scoping review protocol of existing body image guidelines for parents of youth

**Madison F. Vani** [ID]*, **Alishba Mansoor**[◉], **Elise Christopoulos**[◉], **Fengyue Xu**[◉],
**Landyn Meadows**[◉], **Catherine M. Sabiston**

Faculty of Kinesiology & Physical Education, University of Toronto, Toronto, Ontario, Canada

[◉] These authors contributed equally to this protocol.
* madison.vani@mail.utoronto.ca

## Abstract

Body image concerns are commonly experienced by youth in sport and contribute to worse sport experiences and dropout. Parents are positioned to ensure the sport environment is safe and positive for their child, but they do not feel equipped to help their children with body image concerns. Providing parents with tools to support their children is needed. To develop parent-focused body image in sport guidelines, it is necessary to comprehensively summarize parent resources for fostering an adaptive body image among youth. This scoping review will synthesize the available guidelines and/or recommendations for parents on youth body image. The objectives are to (1) identify how many guidelines/recommendations exist for parents to support youth body image; (2) illustrate the guidelines/recommendations' characteristics (i.e., resource type/format; method(s) used to develop resource); and (3) describe the topics/content covered in the guidelines/recommendations. An eight-step methodological framework including parent consultations will inform the review process. The protocol has been registered (https://osf.io/9hq78). Nine databases (Embase, PsycINFO, Social Work Abstracts, and MEDLINE via Ovid, CINAHL, Gender Studies, and SportDiscus via EBSCO, and ERIC and Sociological Abstracts via ProQuest) were searched from January 1, 2014 to February 11, 2025. Inclusion criteria are: qualitative, quantitative, mixed methods, and review articles; written in English; and include recommendations or guidelines developed for parents of children aged 4–18 years and focused on at least one aspect of body image. Titles, abstracts, and full-text articles were screened in duplicate using eligibility criteria. Extracted data will be analyzed using a descriptive numerical summary and conventional content analysis. Institutional research ethics board approval was obtained for parent consultations. The consultations will be integrated with the results and disseminated through academic conferences and a peer-reviewed publication. Findings will also inform the development of parent-focused guidelines for youth body image in sport.

**Data availability statement:** No datasets were generated or analysed during the current study. All relevant data from this study will be made available upon study completion.

**Funding:** Funding for this project was provided through a Social Sciences and Humanities Research Council (https://www.sshrc-crsh. gc.ca/home-accueil-eng.aspx) Partnership Grant awarded to CMS (#895-2023-1007). The funders had no role in study design, decision to publish, or preparation of the manuscript for publication.

**Competing interests:** The authors have declared that no competing interests exist.

## Introduction

Body image is a multidimensional construct involving how one sees, thinks, feels, and behaves in relation to their body's appearance and function [1]. Body image concerns are commonly experienced by youth in sport [2–4] and are associated with lower sport commitment and enjoyment [5]. In addition, concerns about the body's appearance or function is commonly described as a reason for dropout from sport [6]. This is unsurprising given that aspects of the sport context can exacerbate body image concerns. Specifically, sport offers an environment that emphasizes evaluation and appearance, requires specific uniforms that often don't fit comfortably and expose the body, encourages body-based comparisons, and provides opportunities for body and weight commentary [2,3,6].

Parents, guardians, and caregivers (here forth termed 'parents') play an important role in their child's sport experience as they are positioned to ensure the sport environment is safe and positive for their child's wellbeing [2]. Aligned with sociocultural perspectives of body image, parents also influence the development and experience of their child's body image [7], with body image concerns emerging in six-year-old children and intensifying in adolescence [8–9]. Parents can inadvertently contribute to their children's body image concerns by engaging in body or weight commentary or body checking behaviours and conveying rules or expectations around eating and exercise [10–11]. In addition, parents often do not create space for youth to feel comfortable sharing their concerns and they report feeling ill-equipped to help their children with body image concerns [2]. As key agents of socialization, parents play a central role in shaping their children's body image. Accordingly, it is essential to equip parents with the proper tools to support them in this role, especially considering both the potential sport benefits [12,13] and the detrimental impact of body image concerns on wellbeing (e.g., body image concerns are related to higher depressive symptoms and disordered eating; [14,15]) for youth. Further, since parents who model self-compassion and healthy behaviours such as intuitive eating and normalize conversations around concerns can increase positive body image [16, 17], identifying resources that can guide parents to engage in supportive communication around body image is critical. However, there are no known tools that exist for parents to support their child's body image in sport contexts.

To develop parent-focused body image in sport guidelines, it is necessary to comprehensively synthesize the available parent resources for fostering an adaptive body image among youth, regardless of context (e.g., not only in sport). One review examining interventions for parents for youth body dissatisfaction has been conducted [18]. However, this systematic review had a narrow scope since it excluded resource formats that did not fit within an intervention (e.g., recommendations). Alternatively, a scoping review method is a knowledge synthesis that allows for exploratory objectives and provides a broader methodological process to capture studies that have developed resources for parents [19]. In addition, the existing review did not explore other body image constructs beyond body dissatisfaction, limiting the synthesis of parent resources designed for broader body image concepts.

## Study purpose and objectives

The purpose of this scoping review is to examine the available guidelines and/or recommendations for parents on youth body image. Three objectives will guide this review: (1) identify how many guidelines/recommendations exist for parents to support youth body image; (2) illustrate the guidelines/recommendations' characteristics (i.e., resource type/format; method(s) used to develop resource); and (3) describe the topics/content covered in the guidelines/recommendations. Findings from this scoping review will offer a valuable synthesis of evidence-based recommendations and/or guidelines that will inform the development of parent guidelines for youth body image in sport. More broadly, the results will guide researchers' efforts to refine or develop evidence-based tools for parents across diverse contexts, ultimately strengthening the effectiveness of future parent-focused resources.

## Materials and methods

The present scoping review will follow Sabiston et al.'s methodological framework for conducting scoping reviews in sport and exercise psychology [19]. Eight steps will be followed using an iterative and reflexive approach: (1) create and consult a stakeholder group, (2) identify the research question(s), (3) identify relevant studies, (4) create and register a protocol, (5) select and screen studies, (6) chart the data, (7) collate, summarize, and report the results, and (8) consult stakeholders and consider implications. The scoping review will be presented following the checklist outlined in the Preferred Reporting Items for Systematic Reviews and Meta-Analyses – Extension for Scoping Reviews (PRISMA-ScR [20]). The PRISMA protocol checklist is provided in S1 Table. The projected completion date of this scoping review is March 2026. The database search was run on February 11, 2025, screening is complete, and data are being extracted (step 6). Participant recruitment for the stakeholder group will occur in February 2026. It is estimated that stakeholder data collection will be completed in February 2026. Finally, it is expected that article data extraction and analysis will be completed in February 2026 and results are expected in March 2026.

### Step 1: Create and consult a stakeholder group

Knowledge users are typically consulted twice during the review process to obtain perspectives on the (1) methods and (2) results [19]. Initial consultations on the methods usually involve gathering feedback on the research questions, search terms, and eligibility criteria [19]. However, due to time constraints, initial consultations with parents were not carried out. Instead, the design of this scoping review was guided by earlier interviews and discussions with key sport stakeholders (i.e., parents, coaches, organizations), as well as insights drawn from previously conducted interviews with parents of youth athletes [2]. These foundational conversations significantly influenced the methodological approach. Further, as detailed in subsequent sections, the research team engaged topic experts and a health sciences librarian to review and refine the proposed methods. In addition, a reaction meeting in the form of two focus groups will be conducted to support interpretation, reporting, and knowledge dissemination (see step 8 for knowledge user recruitment and reaction meeting details).

### Step 2: Identify the research question(s)

This scoping review's purpose is to examine the available guidelines and/or recommendations for parents on youth body image. Three research questions will guide this review:

(1) How many guidelines and/or recommendations exist for parents to support youth body image?

(2) What are the characteristics (i.e., resource type/format; method(s) used to develop resource) of the guidelines and/or recommendations for parents to support youth body image?

(3) What are the topics/content covered in the guidelines and/or recommendations for parents to support youth body image?

The research questions were developed using the PCC mnemonic [21] to explore the *population* of parents and the *concept* of body image within the *context* of parent-focused guidelines/recommendations. Details on each component are provided below.

**Population.** The population will involve parents, defined as parents, caregivers, and guardians of youth aged 4–18 years. The term youth was used to denote children and adolescents. The minimum age of four years was chosen to ensure the inclusion of preventative resources, given evidence that body image concerns can emerge in early childhood, with reports as young as six years [8]. The upper age boundary was set at 18 years to correspond with widely accepted psychological definitions of adolescence [22]. Articles will be included if the recommendation or guideline was created for parents and focuses on youth body image, regardless of additional identity factors (e.g., gender identity, racial identity, weight status).

**Concept.** Body image is defined as a multidimensional construct involving perceptions, cognitions, affect, and behaviours in relation to the body's appearance and function [1]. Given the inclusive definition, guidelines or recommendations that involve at least one aspect of body image (e.g., body commentary, body appreciation, body dissatisfaction) will be included.

**Context.** The context is defined as scholarly resources including recommendations and/or guidelines that were developed for parents regarding youth body image. Other synonyms of guidelines and recommendations (e.g., toolkit, guide, resource, framework) will also be considered relevant for this scoping review.

**Step 3: Identify relevant studies**

**Information sources.** The following nine electronic databases were searched: Embase, PsycINFO, Social Work Abstracts, and MEDLINE accessed via Ovid, CINAHL, Gender Studies, and SportDiscus accessed via EBSCO, and ERIC and Sociological Abstracts accessed via ProQuest. Journals and reference lists of included sources were hand searched.

**Search strategy.** The research team searched for initial terms related to the population and concept in two databases and identified keywords and index terms. Appropriate controlled terms (e.g., MeSH in MEDLINE and PsycINFO, EMTREE in Embase, CINHAL Headings in CINAHL, Gender Studies Headings in Gender Studies), Boolean logic and operators, and free-text terms were used. The search strategy was refined based on feedback from topic experts and a health sciences librarian. Topic experts in body image and/or review methodologies ($n = 5$) were identified through the research team's networks and consulted to peer review the complete search strategy. Feedback was provided using the CADTH Peer Review Checklist for Search Strategies to enhance rigor and ensure accurate translation of the search strategy across databases [23]. The MEDLINE search strategy is presented in S2 Table. The complete search was conducted across all databases on February 11, 2025. An updated search will be conducted prior to data synthesis to capture any newly published resources and ensure the review is current at the time of reporting.

**Inclusion/exclusion criteria.** The following inclusion criteria were used: (a) qualitative, quantitative, mixed methods, and review articles (e.g., systematic/scoping reviews and meta-analyses); (b) written in English; (c) published between January 1, 2014 and February 11, 2025; (d) include recommendations or guidelines; (e) the recommendations or guidelines were developed for parents of children aged 4–18 years (the resource does not need to cover the entire age range); and (f) the recommendations or guidelines must be focused on at least one aspect of body image (e.g., body/weight commentary, body appreciation, body confidence). Exclusion criteria include: (a) editorials, conference or meeting abstracts and proceedings, preprints, protocols, theses and dissertations, commentaries, policies, organizational briefs, and book chapters; (b) suggestions for parents around supporting youth's body image without explicitly including developed guidelines or recommendations; (c) guidelines or recommendations focused on similar constructs (e.g., self-esteem) to body image without an inclusion of body image concepts; and (d) clinical or treatment-seeking populations (e.g., cancer, clinical diagnosis of mental illness, medical weight management). Subclinical populations (e.g., individuals

presenting with disordered eating or body dysmorphia symptoms without a clinical diagnosis) will not be excluded. Eligibility criteria may be amended through the review process and will be reported in the final publication.

### Step 4: Create and register a protocol

The protocol was prospectively registered with Open Science Framework (https://osf.io/9hq78) on February 11, 2025. The registration includes all necessary information as outlined in the methodological framework [19] and the PRISMA-P checklist (S1 Table; [24]).

### Step 5: Select and screen studies

The first author (MFV) performed the search and search results were exported to Covidence [25] where duplicates were removed. Using eligibility criteria, two stages of study selection were conducted in duplicate by four independent reviewers: (1) title and abstract review and (2) full-text review. Any disagreements were resolved through regular consensus meetings between reviewers, or if needed, two additional reviewers from the research team were consulted.

### Step 6: Chart the data

Aligned with the research questions, a chart form was developed (Table 1), which will be used to provide a descriptive summary of the results. Four reviewers will independently extract relevant data in duplicate from the first five sources as a pilot exercise to ensure consistency. They will meet to consider any discrepancies and will either reach a consensus or consult an additional reviewer who will make a final decision. Formal data extraction from the remaining sources will be conducted by the four reviewers in duplicate, with discrepancies resolved by the research team. An iterative process will occur wherein the chart form may be revised throughout the review process as needed [19].

### Step 7: Collate, summarize, and report the results

Data will be analyzed using a descriptive numerical summary and conventional content analysis [26]. Frequencies will be used to offer a descriptive overview of the sources and results. One member of the review team will conduct the conventional content analysis. Specifically, they will read and re-read the included articles, develop preliminary codes with assistance from the chart form, and create categories based on relationships between codes. The review team will discuss results and team members not involved in the content analysis will act as critical friends [27]. The scoping review's research questions will guide the analyses, wherein results will be discussed in relation to (1) how many recommendations/guidelines are available, (2) the resource characteristics, and (3) the body image topics/content that inform them. The analyses will also summarize findings related to study, population, concept, and context characteristics, and any outcomes reported. The PRISMA-ScR Checklist and knowledge-user consultations will inform the reporting of results, wherein a PRISMA flow diagram will be presented, and findings will be reported descriptively, thematically, and in summary tables [20].

Table 1. Preliminary chart form.

| Study Details (e.g., Authors, Year, Location, Journal) | Study Purpose (i.e., purpose, research questions, and/ or objectives) | Study Design (i.e., quantitative, qualitative, mixed methods, review) | Type of Resource (e.g., guideline, recommendations) | Topic(s) Included (e.g., general body image, body talk) | Method for Developing Resource (e.g., review, Delphi, interviews) | Descriptive Information of Parents (e.g., sex, gender identity, age) | Identity Factors of Youth (e.g., age, gender identity, culture) | Outcomes (i.e., any outcomes measured or described) |
|---|---|---|---|---|---|---|---|---|
|  |  |  |  |  |  |  |  |  |

## Step 8: Consult stakeholders and consider implications

Consulting knowledge users is critical for ensuring that review findings reflect end-users' lived experiences, thereby improving the relevance, applicability, and real-world usefulness of the results [19]. Discussions with parents will help identify the most relevant content and contextual factors needed within resources that may not be fully captured in existing recommendations and guidelines. Their contributions will also support the interpretation of review findings, identify gaps between evidence and lived experience, and strengthen the development of user-centered resources, ultimately enhancing the scoping review's value for guiding future research and resource development.

Institutional research ethics board approval was obtained for knowledge-user engagement. University of Toronto Research Ethics Board approved the parent consultations (Protocol #: 47474). Parents of children aged 4–18 years will be consulted to offer lived experience expertise that will be integrated in the results and interpretation. Parents ($N = 12$–20) will be recruited in February 2026 using social media, existing lists of previous participants who consented to future research, sport organizations' newsletters or listserv, and snowball sampling. Parents will be eligible if they are: (a) 18 years or older, (b) have a 4–18-year-old child in sport (at any level); and (c) able to read, speak, and understand English. Parents will be eligible regardless of geographical location. All participants will complete written informed consent prior to participation in a reaction meeting.

Two focus groups will be conducted online with 6–10 participants per focus group ($n = 12$–20; [28]). Using a semi-structured focus group interview guide, the focus groups will be designed to gain perspectives and feedback on the scoping review results and discuss implications, including how the results can shape the development of a parent guideline for youth body image in sport. Questions will explore parents' overall impressions (e.g., "*Which results were most interesting or surprising to you?*"), result-specific perspectives (e.g., "*Do the topics identified in the resources resonate with you and your experiences with your child?*"), future research ideas (e.g., "*Is there anything missing from the existing parent resources that require us to do more research on?*"), and knowledge dissemination feedback (e.g., "*Who would you share the results of this scoping review with?*"). Focus groups will last 60–90 minutes and will be conducted by two members of the research team via online teleconferencing. During this meeting, the knowledge-users may inform dissemination strategies (e.g., social media posts) with an aim of reaching varied parent and expert groups. In addition, in consultation with knowledge-users, implications regarding future research and practice will be developed.

To analyze focus group data, the discussions will be audio recorded, transcribed verbatim, and a conventional content analysis [26] will be used. One member of the review team will read the transcripts, develop codes, and create categories based on relationships between codes using NVivo software. Results will be discussed among team members and the remaining team members will act as critical friends [27]. Parent consultation results will be integrated with scoping review findings.

## Discussion and conclusion

This scoping review will provide a comprehensive synthesis of available guidelines and/or recommendations for parents on youth body image, including their content and development methods. It will also identify areas where additional evidence-based guidance for parents is needed. Consulting parents as knowledge users will strengthen the practical application of the results by ensuring the findings reflect lived experiences. Parent input will be integrated with the review results to identify priority content, implementation barriers, and strategies for developing user-informed resources that are meaningful to parents.

An integration of the results and parent consultations will be disseminated through conference presentations, a peer-reviewed publication, and infographics on the research lab's website and social media page. In addition, findings will be discussed with parents and youth in the co-design of a guideline for parents on youth body image in sport. Other dissemination strategies may be identified through the parent focus groups. By combining a rigorous synthesis with end-user input, this scoping review will produce actionable guidance that can strengthen parent resources and promote adaptive body image for youth in sport and beyond.

## Supporting information

**S1 Table. PRISMA-P (Preferred Reporting Items for Systematic review and Meta-Analysis Protocols) 2015 checklist.**
(DOCX)

**S2 Table. Preliminary MEDLINE database search strategy.**
(DOCX)

## Acknowledgments

We would like to acknowledge Erica Nekolaichuk, Health Sciences Librarian at the University of Toronto, for their expertise in informing the search strategy. MFV is supported by a Mitacs Accelerate Postdoctoral Fellowship and was supported by a Social Sciences and Humanities Research Council and a Sport Participation Research Initiative supplement during manuscript preparation. CMS holds a Canada Research Chair in Physical Activity and Psychosocial Well-being.

## Author contributions

**Conceptualization:** Madison Vani, Catherine M. Sabiston.

**Funding acquisition:** Catherine M. Sabiston.

**Investigation:** Madison Vani, Alishba Mansoor, Elise Christopoulos, Fengyue Xu, Landyn Meadows.

**Methodology:** Madison Vani, Alishba Mansoor, Elise Christopoulos, Fengyue Xu, Landyn Meadows, Catherine M. Sabiston.

**Project administration:** Madison Vani.

**Supervision:** Madison Vani, Catherine M. Sabiston.

**Visualization:** Madison Vani.

**Writing – original draft:** Madison Vani.

**Writing – review & editing:** Madison Vani, Alishba Mansoor, Elise Christopoulos, Fengyue Xu, Landyn Meadows, Catherine M. Sabiston.

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
