## [Decision Letter · Decision Letter 0]

19 Jan 2026

PLOS One

Dear Dr. Vani,

Thank you for submitting your manuscript to PLOS ONE. After careful consideration, we feel that it has merit but does not fully meet PLOS ONE’s publication criteria as it currently stands. Therefore, we invite you to submit a revised version of the manuscript that addresses the points raised during the review process.

Overall, this manuscript is well-written and presents clear objectives. However, there are a few areas that could benefit from further clarification and additional detail:

Consider strengthening the rationale for the review to provide a clearer foundation for its importance and relevance.The age range of youth is cited as 4-18 years. It would be helpful to include a rationale and supporting citation for this specific range, especially given that <15 years are typically classified as children.Will the review take into account policies or organizational briefs that report recommendations and guidelines? Clarifying this could provide important context for the findings.Regarding Step 8, could you elaborate on the geographical context of the participants? Are you considering any specific settings or regions, and how might this influence the results?Will the Focus Group Discussions (FGDs) be conducted online or in-person? Additionally, could you provide examples of the questions you plan to ask the parents during these discussions?Consider providing a stronger rationale for including FGDs in the review methodology. How do you envision FGDs contributing to the scoping review, and what value will they add?It would be beneficial to add a brief section on the discussion, highlighting the potential implications of the review and outlining a knowledge translation plan.

Please submit your revised manuscript by Mar 05 2026 11:59PM. If you will need more time than this to complete your revisions, please reply to this message or contact the journal office at plosone@plos.org . Please include the following items when submitting your revised manuscript:

We look forward to receiving your revised manuscript.

Kind regards,

Saima Hirani, PhD

Academic Editor

PLOS One

Journal Requirements:

Reviewers' comments:

Reviewer's Responses to Questions

**Comments to the Author**

1. Does the manuscript provide a valid rationale for the proposed study, with clearly identified and justified research questions?

Reviewer #1: Yes

Reviewer #2: No

2. Is the protocol technically sound and planned in a manner that will lead to a meaningful outcome and allow testing the stated hypotheses?

Reviewer #1: Yes

Reviewer #2: Partly

3. Is the methodology feasible and described in sufficient detail to allow the work to be replicable?

Reviewer #1: Yes

Reviewer #2: Yes

4. Have the authors described where all data underlying the findings will be made available when the study is complete?

Reviewer #1: Yes

Reviewer #2: Yes

5. Is the manuscript presented in an intelligible fashion and written in standard English?

Reviewer #1: Yes

Reviewer #2: Yes

You may also provide optional suggestions and comments to authors that they might find helpful in planning their study.

Reviewer #1: Major Criteria Assessment

The protocol meets all the core requirements for a Study Protocol submission to PLOS ONE:

• The background clearly establishes the need for parent-focused body image support tools. The three scoping review objectives are clearly stated and appropriate for the intended purpose of summarizing available resources.

• The protocol follows an established, eight-step methodological framework adapted for scoping reviews in sport and exercise psychology. The search strategy involving consultation with a health sciences librarian and duplicate screening/extraction steps demonstrates strong rigor.

• methods, including the databases searched and the provided search strategy (S2 Table), are detailed enough for reproduction. The plan for data extraction and content analysis (including 'critical friends' for trustworthiness) is explicitly outlined.

• The authors confirm that while screening and data extraction are in progress, the final step, recruitment and consultation with knowledge-users is not complete, confirming the study is not finished. The manuscript title contains the word "Protocol."

• The authors state that no data were generated for the protocol and that all data underlying the eventual findings will be made available upon study completion.

• The PRISMA-ScR checklist is provided as S1 Table and the protocol is prospectively registered with the Open Science Framework (OSF), which is appropriate for this type of review. Ethical approval for the subsequent focus groups is also confirmed.

Minor Clarifications Requested (Required Revisions)

1. Scope Clarification on Clinical Populations (Exclusion Criteria):

• Exclusion criterion (d) states that the protocol will exclude "clinical or treatment-seeking populations (e.g., cancer, clinical diagnosis of mental illness, medical weight management)."

• Action Required: Please clarify if this exclusion strictly applies to clinically diagnosed conditions or if it also excludes resources for parents of youth with subclinical disordered eating or body dysmorphia (which are mentioned as related conditions in the Introduction). Given the close relationship between body image concerns and disordered eating, clarifying this boundary will ensure the scope is precisely defined and avoid ambiguity during the selection process.

2. Stakeholder Consultation Rationale (Step 1):

• The authors note that the initial formal consultation (Step 1 in the cited framework) was not carried out due to time constraints, but this step was addressed by prior semi-structured interviews that informed the study design.

• Action Required: Please briefly adjust the text under the "Stakeholder Consultation" heading to state explicitly that the findings from the prior semi-structured interviews served as the initial stakeholder consultation, justifying the omission of the separate formal consultation meeting. This strengthens the integrity of following the adapted methodology.

3. Search Update:

• The protocol states that the search was conducted in January 2024. For a protocol that is currently undergoing review, it is often best practice to plan a final update search closer to the time of data analysis completion.

• Action Required: Please add a brief sentence to the "Search Strategy" section confirming that a final, minor update search will be conducted prior to data synthesis to capture any very recently published guidelines, ensuring the review is as current as possible upon publication of the final report.

Reviewer #2: rationale of doing this scoping review is not very clear .There are so many factor that might influence body -image of sports persons/athletes.

**Do you want your identity to be public for this peer review?** For information about this choice, including consent withdrawal, please see our Privacy Policy

Reviewer #1:**Yes:**

Reviewer #2: No

---

## [Author Response · Author response to Decision Letter 1]

3 Feb 2026

To the Academic Editor and Reviewers,

We are grateful for your feedback, and for the opportunity to revise and resubmit this work. We have completed a careful revision based on your recommendations. Specific comments by the reviewers can be found in bolded text with our responses in non-bolded text (text referenced in the paper is in italics). We are very much looking forward to your feedback and hope that you will find our manuscript suitable for publication in PLOS One.

ACADEMIC EDITOR COMMENTS

Overall, this manuscript is well-written and presents clear objectives. However, there are a few areas that could benefit from further clarification and additional detail:

1. Consider strengthening the rationale for the review to provide a clearer foundation for its importance and relevance.

The rationale for the review was strengthened in the manuscript (pp. 3-4):

Parents, guardians, and caregivers (here forth termed ‘parents’) play an important role in their child’s sport experience as they are positioned to ensure the sport environment is safe and positive for their child’s wellbeing [2]. Aligned with sociocultural perspectives of body image, parents also influence the development and experience of their child’s body image [7], with body image concerns emerging in six-year-old children and intensifying in adolescence [8-9]. Parents can inadvertently contribute to their children’s body image concerns by engaging in body or weight commentary or body checking behaviours and conveying rules or expectations around eating and exercise [10-11]. In addition, parents often do not create space for youth to feel comfortable sharing their concerns and they report feeling ill-equipped to help their children with body image concerns [2]. As key agents of socialization, parents play a central role in shaping their children’s body image. Accordingly, it is essential to equip parents with the proper tools to support them in this role, especially considering both the potential sport benefits [12,13] and the detrimental impact of body image concerns on wellbeing (e.g., body image concerns are related to higher depressive symptoms and disordered eating; [14,15]) for youth. Further, since parents who model self-compassion and healthy behaviours such as intuitive eating and normalize conversations around concerns can increase positive body image [16,17], identifying resources that can guide parents to engage in supportive communication around body image is critical. However, there are no known tools that exist for parents to support their child’s body image in sport contexts.

To develop parent-focused body image in sport guidelines, it is necessary to comprehensively synthesize the available parent resources for fostering an adaptive body image among youth, regardless of context (e.g., not only in sport). One review examining interventions for parents for youth body dissatisfaction has been conducted [18]. However, this systematic review had a narrow scope since it excluded resource formats that did not fit within an intervention (e.g., recommendations). Alternatively, a scoping review method is a knowledge synthesis that allows for exploratory objectives and provides a broader methodological process to capture studies that have developed resources for parents [19]. In addition, the existing review did not explore other body image constructs beyond body dissatisfaction, limiting the synthesis of parent resources designed for broader body image concepts.

More broadly, the results will guide researchers’ efforts to refine or develop evidence-based tools for parents across diverse contexts, ultimately strengthening the effectiveness of future parent-focused resources.

Tiggemann, M. (2011). Sociocultural perspectives on human appearance and body image. In T. F. Cash & L. Smolak (Eds.), Body image: A handbook of science, practice, and prevention (2nd ed., pp. 12–19). The Guilford Press.

Lowes, J., & Tiggemann, M. (2003). Body dissatisfaction, dieting awareness and the impact of parental influence in young children. British Journal of Health Psychology, 8(2), 135–147. https://doi.org/10.1348/135910703321649123

Voelker, D. K., Reel, J. J., & Greenleaf, C. (2015). Weight status and body image perceptions in adolescents: Current perspectives. Adolescent Health, Medicine and Therapeutics, 149-158. https://doi.org/10.2147/AHMT.S68344

Rodgers, R., & Chabrol, H. (2009). Parental attitudes, body image disturbance and disordered eating amongst adolescents and young adults: A review. European Eating Disorders Review: The Professional Journal of the Eating Disorders Association, 17(2), 137–151. https://doi.org/10.1002/erv.907

Tremblay, L., & Limbos, M. (2009). Body image disturbance and psychopathology in children: Research evidence and implications for prevention and treatment. Current Psychiatry Reviews, 5(1), 62–72. https://doi.org/10.2174/157340009787315307

Goslin, A., & Koons-Beauchamp, D. (2023). The mother-daughter relationship and daughter's positive body image: A systematic review. The Family Journal, 31(1), 128–139. https://doi.org/10.1177/10664807221104109

Rodgers, R. F., Gordon, A. R., Burke, N. L., & Ciao, A. (2024). Parents and caregivers as key players in the prevention and identification of body image concerns and eating disorders among early adolescents. Eating Disorders, 32(6), 703–726. https://doi.org/10.1080/10640266.2024.2366546

Hart, L. M., Cornell, C., Damiano, S. R., & Paxton, S. J. (2015). Parents and prevention: A systematic review of interventions involving parents that aim to prevent body dissatisfaction or eating disorders. International Journal of Eating Disorders, 48(2), 157–169. https://doi.org/10.1002/eat.22284

2. The age range of youth is cited as 4-18 years. It would be helpful to include a rationale and supporting citation for this specific range, especially given that <15 years are typically classified as children.

A rationale for age range including citations was provided (p. 6):

The term youth was used to denote children and adolescents. The minimum age of four years was chosen to ensure the inclusion of preventative resources, given evidence that body image concerns can emerge in early childhood, with reports as young as six years [8]. The upper age boundary was set at 18 years to correspond with widely accepted psychological definitions of adolescence [22].

Lowes, J., & Tiggemann, M. (2003). Body dissatisfaction, dieting awareness and the impact of parental influence in young children. British Journal of Health Psychology, 8(2), 135–147. https://doi.org/10.1348/135910703321649123

American Psychological Association. (2002). Developing adolescents: A reference for professionals. https://www.apa.org/topics/teens/developing-adolescents-professionals-reference

3. Will the review take into account policies or organizational briefs that report recommendations and guidelines? Clarifying this could provide important context for the findings.

The review will not include policies or organizational briefs as our focus for this review was on peer-reviewed literature. We have clarified this by including these in the exclusion criteria (p. 8):

Exclusion criteria include: (a) editorials, conference or meeting abstracts and proceedings, preprints, protocols, theses and dissertations, commentaries, policies, organizational briefs, and book chapters…

4. Regarding Step 8, could you elaborate on the geographical context of the participants? Are you considering any specific settings or regions, and how might this influence the results?

Geographical context was not included as an inclusion criterion since parents will be eligible regardless of location. We acknowledge that regional differences in sociocultural context and resource availability may influence the perspectives parents provide. This will be considered when interpreting the data and writing the results and discussion of the scoping review. The geographical context of participants was clarified in the manuscript (p. 11):

Parents will be eligible regardless of geographical location.

5. Will the Focus Group Discussions (FGDs) be conducted online or in-person? Additionally, could you provide examples of the questions you plan to ask the parents during these discussions?

Focus groups will be conducted online. This information is provided in the manuscript (p. 11):

Two focus groups will be conducted online with 6-10 participants per focus group.

Focus groups will last 60-90 minutes and will be conducted by two members of the research team via online teleconferencing.

We have provided examples of the questions we plan to ask in the focus groups (p. 11):

Questions will explore parents’ overall impressions (e.g., “Which results were most interesting or surprising to you?”), result-specific perspectives (e.g., “Do the topics identified in the resources resonate with you and your experiences with your child?”), future research ideas (e.g., “Is there anything missing from the existing parent resources that require us to do more research on?”), and knowledge dissemination feedback (e.g., “Who would you share the results of this scoping review with?”).

6. Consider providing a stronger rationale for including FGDs in the review methodology. How do you envision FGDs contributing to the scoping review, and what value will they add?

A stronger rationale for end-user engagement was provided (p. 10):

Consulting knowledge users is critical for ensuring that review findings reflect end-users’ lived experiences, thereby improving the relevance, applicability, and real-world usefulness of the results [19]. Discussions with parents will help identify the most relevant content and contextual factors needed within resources that may not be fully captured in existing recommendations and guidelines. Their contributions will also support the interpretation of review findings, identify gaps between evidence and lived experience, and strengthen the development of user-centered resources, ultimately enhancing the scoping review’s value for guiding future research and resource development.

7. It would be beneficial to add a brief section on the discussion, highlighting the potential implications of the review and outlining a knowledge translation plan.

A discussion section was added and integrated with the conclusion (p. 12):

This scoping review will provide a comprehensive synthesis of available guidelines and/or recommendations for parents on youth body image, including their content and development methods. It will also identify areas where additional evidence-based guidance for parents is needed. Consulting parents as knowledge users will strengthen the practical application of the results by ensuring the findings reflect lived experiences. Parent input will be integrated with the review results to identify priority content, implementation barriers, and strategies for developing user-informed resources that are meaningful to parents.

An integration of the results and parent consultations will be disseminated through conference presentations, a peer-reviewed publication, and infographics on the research lab’s website and social media page. In addition, findings will be discussed with parents and youth in the co-design of a guideline for parents on youth body image in sport. Other dissemination strategies may be identified through the parent focus groups. By combining a rigorous synthesis with end-user input, this scoping review will produce actionable guidance that can strengthen parent resources and promote adaptive body image for youth in sport and beyond.

REVIEWER 1 COMMENTS

The study presents a technically sound and well-justified scoping review protocol. The rationale for synthesizing existing body image resources for parents, especially to inform the development of future sport-specific guidelines, is compelling and addresses a relevant gap in the literature. The methods are described in sufficient detail to ensure replicability and rigor. I recommend the protocol for Acceptance with Minor Revisions to address a few points of clarification regarding the scope and stakeholder engagement.

Major Criteria Assessment

The protocol meets all the core requirements for a Study Protocol submission to PLOS ONE:

• The background clearly establishes the need for parent-focused body image support tools. The three scoping review objectives are clearly stated and appropriate for the intended purpose of summarizing available resources.

• The protocol follows an established, eight-step methodological framework adapted for scoping reviews in sport and exercise psychology. The search strategy involving consultation with a health sciences librarian and duplicate screening/extraction steps demonstrates strong rigor.

• methods, including the databases searched and the provided search strategy (S2 Table), are detailed enough for reproduction. The plan for data extraction and content analysis (including 'critical friends' for trustworthiness) is explicitly outlined.

• The authors confirm that while screening and data extraction are in progress, the final step, recruitment and consultation with knowledge-users is not complete, confirming the study is not finished. The manuscript title contains the word "Protocol."

• The authors state that no data were generated for the protocol and that all data underlying the eventual findings will be made available upon study completion.

• The PRISMA-ScR checklist is provided as S1 Table and the protocol is prospectively registered with the Open Science Framework (OSF), which is appropriate for this type of review. Ethical approval for the subsequent focus groups is also confirmed.

We thank Reviewer 1 for their careful review and thoughtful comments.

Minor Clarifications Requested (Required Revisions)

1. Scope Clarification on Clinical Populations (Exclusion Criteria):

• Exclusion criterion (d) states that the protocol will exclude "clinical or treatment-seeking populations (e.g., cancer, clinical diagnosis of mental illness, medical weight management)."

• Action Required: Please clarify if this exclusion strictly applies to clinically diagnosed conditions or if it also excludes resources for parents of youth with subclinical disordered eating or body dysmorphia (which are mentioned as related conditions in the Introduction). Given the close relationship between body image concerns and disordered eating, clarifying this boundary will ensure the scope is precisely defined and avoid ambiguity during the selection process.

Thank you for identifying this need for clarification. We have clarified this exclusion criteria in the manuscript (p. 8):

Subclinical populations (e.g., individuals presenting with disordered eating or body dysmorphia symptoms without a clinical diagnosis) will not be excluded.

2. Stakeholder Consultation Rationale (Step 1):

• The authors note that the initial formal consultation (Step 1 in the cited framework) was not carried out due to time constraints, but this step was addressed by prior semi-structured interviews that informed the study design.

• Action Required: Please briefly adjust the text under the "Stakeholder Consultation" heading to state explicitly that the findings from the prior semi-structured interviews served as the initial stakeholder consultation, justifying the omission of the separate formal consultation meeting. This strengthens the integrity of following the adapted methodology.

Thank you for this comment. We did not revise the text to state that the prior semi-structured interviews served as the initial stakeholder consultation. According to the scoping review methodology followed (Sabiston et al., 2022), the purpose of the initial consultation is to obtain stakeholder feedback on the review methods (e.g., research questions, search terms, eligibility criteria). While the prior interviews with parents informed the rationale for conducting the review and shaped its overall focus, they did not involve consultation on the developed scoping review methods. Therefore, in this case, we did not consider it methodologically appropriate to characterize these interviews as fulfilling the initial stakeholder consultation step. We did however ad

---

## [Editor Report · Decision Letter 1]

5 Feb 2026

A scoping review protocol of existing body image guidelines for parents of youth

PONE-D-25-25906R1

Dear Dr. Madison Vani,

We’re pleased to inform you that your manuscript has been judged scientifically suitable for publication and will be formally accepted for publication once it meets all outstanding technical requirements.

Kind regards,

Saima Hirani, PhD RN

Academic Editor

PLOS One

---

## [Editor Report · Acceptance letter]

PONE-D-25-25906R1

PLOS One

Dear Dr. Vani,

I'm pleased to inform you that your manuscript has been deemed suitable for publication in PLOS One. Congratulations! Your manuscript is now being handed over to our production team.

Kind regards,

on behalf of

Dr. Saima Hirani

Academic Editor

PLOS One